# GAUDI: A Neural Architect for Immersive 3D Scene Generation

**Miguel Angel Bautista**∗    **Pengsheng Guo**∗    **Samira Abnar**    **Walter Talbott**

**Alexander Toshev**    **Zhuoyuan Chen**    **Laurent Dinh**    **Shuangfei Zhai**    **Hanlin Goh**

**Daniel Ulbricht**    **Afshin Dehghan**    **Josh Susskind**

Apple
https://github.com/apple/ml-gaudi

## Abstract

We introduce **GAUDI**, a generative model capable of capturing the distribution of complex and realistic 3D scenes that can be rendered immersively from a moving camera. We tackle this challenging problem with a scalable yet powerful approach, where we first optimize a latent representation that disentangles radiance fields and camera poses. This latent representation is then used to learn a generative model that enables both unconditional and conditional generation of 3D scenes. Our model generalizes previous works that focus on single objects by removing the assumption that the camera pose distribution can be shared across samples. We show that GAUDI obtains state-of-the-art performance in the unconditional generative setting across multiple datasets and allows for conditional generation of 3D scenes given conditioning variables like sparse image observations or text that describes the scene.

## 1   Introduction

In order for learning systems to be able to understand and create 3D spaces, progress in generative models for 3D is sorely needed. The quote *"The creation continues incessantly through the media of man."* is often attributed to *Antoni Gaudí*, who we pay homage to with our method's name. In this paper we ask the question: *can creation continue through the media of learning machines?* We are interested in generative models that can capture the distribution of 3D scenes and then render views from scenes sampled from the learned distribution. Extensions of such generative models to conditional inference problems could have tremendous impact in a wide range of tasks in machine learning and computer vision. For example, one could sample plausible scene completions that are consistent with an image observation, or a text description (see Fig. 1 for 3D scenes sampled from GAUDI). In addition, such models would be of great practical use in model-based reinforcement learning and planning [14], SLAM [43], or 3D content creation.

Recent works on generative modeling for 3D objects or scenes [60, 6, 8] employ a Generative Adversarial Network (GAN) where the generator explicitly encodes radiance fields — a parametric function that takes as input the coordinates of a point in 3D space and camera pose, and outputs a density scalar and RGB value for that 3D point. Images can be rendered from the radiance field generated by the model by passing the queried 3D points through the volume rendering equation

---

∗ denotes equal contribution. Corresponding email: mbautistamartin@apple.com

36th Conference on Neural Information Processing Systems (NeurIPS 2022).

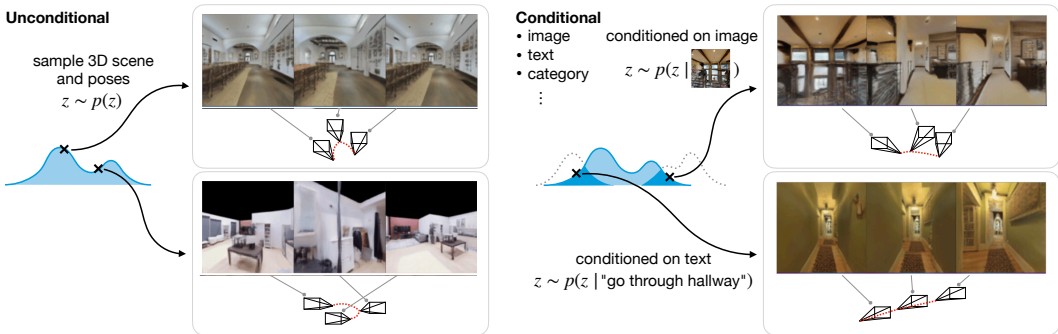

Figure 1: GAUDI allows to model both conditional and unconditional distributions over complex 3D scenes. Sampled scenes and poses from (left) the unconditional distribution, and (right) a distribution conditioned on an image observation or a text prompt.

to project onto any 2D camera view. While compelling on small or simple 3D datasets (*e.g.* single objects or a small number of indoor scenes), GANs suffer from training pathologies including mode collapse [58, 65] and are difficult to train on data for which a canonical coordinate system does not exist, as is the case for 3D scenes [61]. In addition, one key difference between modeling distributions of 3D *objects vs. scenes* is that when modeling objects, the distribution of valid camera poses does not depend on each object and is defined per dataset (*i.e.* typically as $SO(3)$), which is not true for scenes. See Fig. 3(b) and note how the two scene layouts define different areas of navigable space (different dark grey shaded areas) where valid camera poses can be placed, revealing a strong dependency between scenes and their camera pose distributions. GAUDI captures this dependency by modeling the joint distribution of scenes and camera poses.

In GAUDI, we map each trajectory (*i.e.* a sequence of posed images from a 3D scene) into a latent representation that encodes a radiance field (*e.g.* the 3D scene) and camera path in a completely disentangled way. We find these latent representations by interpreting them as free parameters and formulating an optimization problem where the latent representation for each trajectory is optimized via a reconstruction objective. This simple training process is scalable to thousands of trajectories. Interpreting the latent representation of each trajectory as a free parameter also makes it simple to handle a large and variable number of views for each trajectory rather than requiring a sophisticated encoder architecture to pool across a large number of views. After optimizing latent representations for an observed empirical distribution of trajectories, we learn a generative model over the set of latent representations. In the unconditional case, the model can sample radiance fields entirely from the prior distribution learned by the model, allowing it to synthesize scenes by interpolating within the latent space. In the conditional case, conditional variables available to the model at training time (*e.g.* images, text prompts, etc.) can be used to generate radiance fields consistent with those variables. Our contributions can be summarized as:

• We scale 3D scene generation to thousands of indoor scenes containing hundreds of thousands of images, without suffering from mode collapse or canonical orientation issues during training.

• We introduce a novel denoising optimization objective to find latent representations that jointly model a radiance field and the camera poses in a disentangled manner.

• Our approach obtains state-of-the-art generation performance across multiple datasets.

• Our approach allows for various generative setups: unconditional generation as well as conditional on images or text.

## 2   Related Work

In recent years the field has witnessed outstanding progress in generative modeling for the 2D image domain, with most approaches focusing either on adversarial [21, 22] or auto-regressive models [69, 46, 11]. More recently, score matching based approaches [18, 62] have gained popularity. In particular, Denoising Diffusion Probabilistic Models (DDPMs) [17, 37, 52, 68] have emerged as strong contenders to both adversarial and auto-regressive approaches. In DDPMs, the goal is to learn

a step-by-step inversion of a fixed diffusion Markov Chain that gradually transforms an empirical data distribution to a fixed posterior, which typically takes the form of an isotropic Gaussian distribution. In parallel, the last couple of years have seen a revolution in how 3D data is represented within neural networks. By representing a 3D scene as a radiance field, NeRF [33] introduces an approach to optimize the weights of a MLP to represent the radiance of 3D points that fall inside the field-of-view of a given set of posed RGB images. Given the radiance for a set of 3D points that lie on a ray shot from a given camera pose, NeRF [33] uses volumetric rendering to compute the color for the corresponding pixel and optimizes the MLP weights via a reconstruction loss in image space.

A few attempts have also been made at incorporating a radiance field representation within generative models. Most approaches have focused on the problem of single objects with known canonical orientations like faces or Shapenet objects with shared camera pose distributions across samples in a dataset [60, 6, 38, 25, 5, 12, 75, 47, 10]. Extending these approaches from single objects to completely unconstrained 3D scenes is an unsolved problem. One paper worth mentioning in this space is GSN [8], which breaks the radiance field into a grid of local radiance fields that collectively represent a scene. While this decomposition of radiance fields endows the model with high representational capacity, GSN still suffers from the standard training pathologies of GANs, like mode collapse [65], which are exacerbated by the fact that unconstrained 3D scenes do not have a canonical orientation. As we show in our experiments (cf. Sect. 4), these issues become prominent as the training set size increases, impacting the capacity of the generative model to capture complex distributions. Separately, a line of recent approaches have also studied the problem of learning generative models of scenes without employing radiance fields [40, 70, 51]. These works assume that the model has access to room layouts and a database of object CAD models during training, simplifying the problem of scene generation to a selection of objects from the database and pose predictions for each object.

Finally, approaches that learn to predict a target view given a single (or multiple) source view and relative pose transformation have been recently proposed [27, 74, 57, 9, 13]. The pure reconstruction objective employed by these approaches forces them to learn a deterministic conditional function that maps a source image and a relative camera transformation to a target image. The first is that this scene completion problem is ill-posed (*e.g.* given a single source view of a scene there are multiple target completions that are equally likely). Attempts at modeling the problem in a probabilistic manner have been proposed [53, 49]. However, these approaches suffer from inconsistency in predicted scenes because they do not explicitly model a 3D consistent representation like a radiance field.

## 3  GAUDI

Our goal is to learn a generative model given an empirical distribution of trajectories over 3D scenes. Let $X = \{x_{i \in \{0,...,n\}}\}$ denote a collection of examples defining an empirical distribution, where each example $x_i$ is a trajectory. Every trajectory $x_i$ is defined as a variable length sequence of corresponding RGB, depth images and 6DOF camera poses (see Fig. 3).

We decompose the task of learning a generative model in two stages. First, we obtain a latent representation $\mathbf{z} = [\mathbf{z}_{\text{scene}}, \mathbf{z}_{\text{pose}}]$ for each example $x \in X$ that represents the scene radiance field and pose in separate disentangled vectors. Second, given a set of latents $Z = \{\mathbf{z}_{i \in \{0,...,n\}}\}$ we learn the distribution $p(Z)$.

### 3.1  Optimizing latent representations for radiance fields and camera poses

We now turn to the task of finding a latent representation $\mathbf{z} \in Z$ for each example $x \in X$ (*i.e.* for each trajectory in the empirical distribution). To obtain this latent representation we take an encoder-less view and interpret $\mathbf{z}$'s as free parameters to be found via an optimization problem [2, 39]. To map latents $\mathbf{z}$ to trajectories $x$, we design a network architecture (*i.e.* a decoder) that disentangles camera poses and radiance field parameterization. Our decoder architecture is composed of 3 networks (shown in Fig. 2):

• The **camera pose decoder** network $c$ (parameterized by $\theta_c$), is responsible for predicting camera poses $\hat{\mathbf{T}}_s \in SE(3)$ at the normalized temporal position $s \in [-1, 1]$ in the trajectory, conditioned on $\mathbf{z}_{\text{pose}}$ which represents the camera poses for the *whole* trajectory. To ensure that the output of $c$ is a valid camera pose (*e.g.* an element of $SE(3)$), we output a 3D vector representing a *normalized* quaternion $\mathbf{q}_s$ for the orientation and a 3D translation vector $\mathbf{t}_s$.

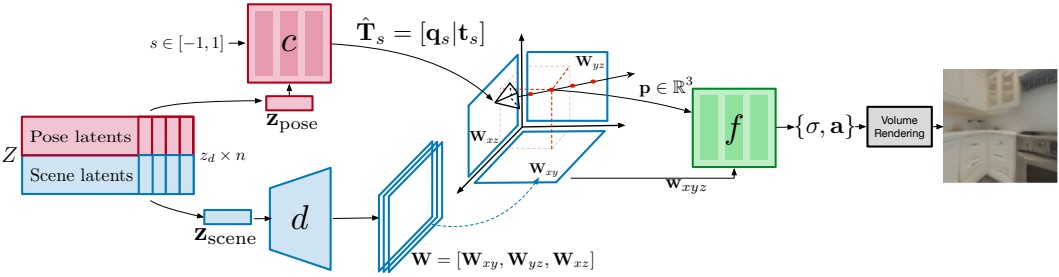

Figure 2: Architecture of the decoder model that disentangles camera poses from 3D geometry and appearance of the scene. Our decoder is composed by 3 submodules. A decoder $d$ that takes as input a latent code representing the scene $\mathbf{z}_{\text{scene}}$ and produces a factorized representation of 3D space via a tri-plane latent encoding $\mathbf{W}$. A radiance field network $f$ that takes as input points $\mathbf{p} \in \mathbf{R}^3$ and is conditioned on $\mathbf{W}$ to predict a density $\sigma$ and a signal $\mathbf{a}$ to be rendered via volumetric rendering (Eq. 1). Finally, we decode the camera poses through a network $c$ that takes as input a normalized temporal position $s \in [-1, 1]$ and is conditioned on $\mathbf{z}_{\text{pose}}$ which represents camera poses for the whole trajectory $x$ to predict the camera pose $\hat{\mathbf{T}}_s \in SE(3)$.

• The **scene decoder** network $d$ (parameterized by $\theta_d$), is responsible for predicting a conditioning variable for the radiance field network $f$. This network takes as input a latent code that represents the scene $\mathbf{z}_{\text{scene}}$ and predicts an axis-aligned tri-plane representation [41, 5] $\mathbf{W} \in \mathbb{R}^{3 \times S \times S \times F}$. Which correspond to 3 feature maps $[\mathbf{W}_{xy}, \mathbf{W}_{xz}, \mathbf{W}_{yz}]$ of spatial dimension $S \times S$ and $F$ channels, one for each axis aligned plane: $xy$, $xz$ and $yz$.

• The **radiance field decoder** network $f$ (parameterized by $\theta_f$), is tasked with reconstructing image level targets using the volumetric rendering equation in Eq. 1. The input to $f$ is $\mathbf{p} \in \mathbb{R}^3$ and the tri-plane representation $\mathbf{W} = [\mathbf{W}_{xy}, \mathbf{W}_{xz}, \mathbf{W}_{yz}]$ (we do not condition on the camera orientation to improve consistency [12]). Given a 3D point $\mathbf{p} = [i, j, k]$ for which radiance is to be predicted, we orthogonally project $\mathbf{p}$ into each plane in $\mathbf{W}$ and perform bi-linear sampling. We concatenate the 3 bi-linearly sampled vectors into $\mathbf{w}_{xyz} = [\mathbf{W}_{xy}(i, j), \mathbf{W}_{xz}(j, k), \mathbf{W}_{yz}(i, k)] \in \mathbb{R}^{3F}$, which is used to condition the radiance field function $f$. We implement $f$ as a MLP that outputs a density value $\sigma$ and a signal $\mathbf{a}$. To predict the value $\mathbf{v}$ of a pixel, the volumetric rendering equation is used (cf. Eq. 1) where a 3D point is expressed as ray direction $\mathbf{r}$ (corresponding with the pixel location) at particular depth $u$.

$$\mathbf{v}(\mathbf{r}, \mathbf{W}) = \int_{u_n}^{u_f} Tr(u)\sigma\left(\mathbf{r}(u), \mathbf{w}_{xyz}\right) \mathbf{a}\left(\mathbf{r}(u), \mathbf{w}_{xyz}\right) du$$

$$Tr(u) = \exp\left(-\int_{u_n}^{u} \sigma(\mathbf{r}(u), \mathbf{w}_{xyz}) du\right). \tag{1}$$

We formulate a reconstruction objective to jointly optimize for $\theta_d$, $\theta_c$, $\theta_f$ and $\{\mathbf{z}\}_{i=\{0,...,n\}}$, shown in Eq. 2. Note that while latents $\mathbf{z}$ are optimized for each example $x$ independently, the parameters of the networks $\theta_d$, $\theta_c$, $\theta_f$ are amortized across all examples $x \in X$. As opposed to previous auto-decoding approaches [2, 39], each latent $\mathbf{z}$ is perturbed during training with additive noise that is proportional to the empirical standard deviation across all latents, $\mathbf{z} = \mathbf{z} + \beta \mathcal{N}(0, \text{std}(Z))$, inducing a contractive representation [50]. In this setting, $\beta$ controls the trade-off between the entropy of the distribution of latents $\mathbf{z} \in Z$ and the reconstruction quality. With $\beta = 0$ the distribution of $\mathbf{z}$'s becomes a set of indicator functions (*i.e.* similar as one would get from a vanilla auto-encoder). For large $\beta > 0$ the structure in distribution of $\mathbf{z}$'s is destroyed, as latents are perturbed with large magnitudes of noise. We use a small $\beta > 0$ value to enforce a latent space in which interpolated samples (or samples that contain small deviations from the empirical distribution, as the ones that one might get from sampling a subsequent generative model) are included in the support of the decoder function, sacrificing a small cost in reconstruction fidelity.

$$\min_{\theta_d, \theta_f, \theta_c, Z} \mathbb{E}_{x \sim X} \left[\mathcal{L}_{\text{scene}}(\mathbf{x}_s^{\text{im}}, \mathbf{z}_{\text{scene}}, \mathbf{T}_s) + \lambda \mathcal{L}_{\text{pose}}(\mathbf{T}_s, \mathbf{z}_{\text{pose}}, s)\right] \tag{2}$$

We optimize parameters $\theta_d, \theta_f, \theta_c$ and latents $\mathbf{z} \in Z$ with two different losses. The first loss function $\mathcal{L}_{\text{scene}}$ measures the reconstruction between the radiance field encoded in $\mathbf{z}_{\text{scene}}$ and the images in

the trajectory $\mathbf{x}_s^{\text{im}}$ (where $s$ denotes the normalized temporal position of the frame in the trajectory), given ground-truth camera poses $\mathbf{T}_s$ required for rendering. We use an $l_2$ loss for RGB and $l_1$ for depth [1]. The second loss function $\mathcal{L}_{\text{pose}}$ measures the camera pose reconstruction error between the poses $\hat{\mathbf{T}}_s$ encoded in $\mathbf{z}_{\text{pose}}$ and the ground-truth poses. We employ an $l_2$ loss on translation and $l_1$ loss for the normalized quaternion part of the camera pose. Although theoretically normalized quaternions are not necessarily unique (*e.g.* $\mathbf{q}$ and $-\mathbf{q}$) we do not observe any issues empirically during training.

## 3.2 Prior Learning

Given a set of latents $\mathbf{z} \in Z$ resulting from minimizing the objective in Eq. 2, our goal is to learn a generative model $p(Z)$ that captures their distribution (*i.e.* after minimizing the objective in Eq. 2 we interpret $\mathbf{z} \in Z$ as examples from an empirical distribution in latent space). In order to model $p(Z)$ we employ a Denoising Diffusion Probabilistic Model (DDPM) [17], a recent score-matching [18] based model that learns to reverse a diffusion Markov Chain with a large but finite number of timesteps. In DDPMs [17] it is shown that this reverse process is equivalent to learning a sequence of denoising auto-encoders with tied weights. The supervised denoising objective in DDPMs makes learning $p(Z)$ simple and scalable. This allows us to learn a powerful generative model that enables both unconditional and conditional generation of 3D scenes. For training our prior $p_{\theta_p}(Z)$ we take the objective function in [17] defined in Eq. 3. In Eq. 3 $t$ denotes the timestep, $\epsilon \sim \mathcal{N}(0, \mathbf{I})$ is the noise and $\bar{\alpha}_t$ is a noise magnitude parameter with a fixed scheduling. Finally, $\epsilon_{\theta_p}$ denotes the denoising model.

$$\min_{\theta_p} \mathbb{E}_{t, \mathbf{z} \sim Z, \epsilon \sim \mathcal{N}(0, \mathbf{I})} \left[ \| \epsilon - \epsilon_{\theta_p}(\sqrt{\bar{\alpha}_t} \mathbf{z} + \sqrt{1 - \bar{\alpha}_t} \epsilon, t) \|^2 \right] \tag{3}$$

At inference time, we sample $\mathbf{z} \sim p_{\theta_p}(Z)$ by following the inference process in DDPMs. We start by sampling $\mathbf{z}_T \sim \mathcal{N}(0, \mathbf{I})$ and iteratively apply $\epsilon_{\theta_p}$ to gradually denoise $\mathbf{z}_T$, thus reversing the diffusion Markov Chain to obtain $\mathbf{z}_0$. We then feed $\mathbf{z}_0$ as input to the decoder architecture (cf. Fig. 2) and reconstruct a radiance field and a camera path.

If the goal is to learn a conditional distribution of the latents $p(Z|Y)$, given paired data $\{\mathbf{z} \in Z, y \in Y\}$, the denoising model $\epsilon_\theta$ is augmented with a conditioning variable $y$, resulting in $\epsilon_{\theta_p}(\mathbf{z}, t, y)$, implementation details about how the conditioning variable is used in the denoising architecture can be found in the appendix C.

## 4 Experiments

In this section we show the applicability of GAUDI to multiple problems. First, we evaluate reconstruction quality and performance of the reconstruction stage. Then, we evaluate the performance of our model in generative tasks including unconditional and conditional inference, in which radiance fields are generated from conditioning variables corresponding to images or text prompts. Full experimental settings and details can be found in the appendix B.

### 4.1 Data

We report results on 4 datasets: Vizdoom [23], Replica [64], VLN-CE [26] and ARKit Scenes [1], which vary in number of scenes and complexity (see Fig. 3 and Tab. 1).

**Vizdoom** [23]: Vizdoom is a synthetic simulated environment with simple texture and geometry. We use the data provided by [8] to train our model. It is the simplest dataset in terms of number of scenes and trajectories, as well as texture, serving as a test bed to examine GAUDI in the simplest setting.

**Replica** [64]: Replica is a dataset comprised of 18 realistic scenes from which trajectories are rendered via Habitat [59]. We used the data provided by [8] to train our model.

**VLN-CE** [26]: VLN-CE is a dataset originally designed for vision and language navigation in continuous environments. This dataset is composed of 3.6K trajectories of an agent navigating between two points in a 3D scene from the Matterport 3D dataset [7]. We render observations via

---

[1] We obtain depth predictions by aggregating densities across a ray as in [33]

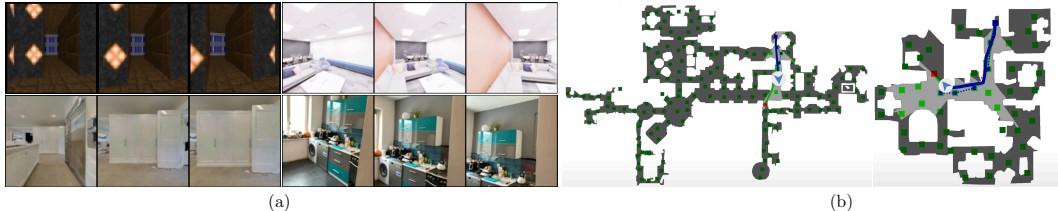

(a)                                                                                    (b)

Figure 3: (a) Examples of the 4 datasets we use in this paper (from left to right): Vizdoom [23], Replica [64], VLN-CE [26], ARKitScenes [1]. (b) Layouts for two scenes in VLN-CE [26], where navigable areas are shaded in dark gray. Blue and red dots represent start-end positions and the camera path is highlighted in blue.

Habitat [59]. Notably, this dataset contains also textual descriptions of the trajectories taken by an agent. In Sect. 4.5 we train GAUDI in a conditional manner to generate 3D scenes given a description.

**ARKitScenes** [1]: ARKitScenes is a dataset of scans of indoor spaces. This dataset contains more than 5K scans of about 1.6K different indoor spaces. As opposed to the previous datasets where RGB, depth and camera poses are obtained via rendering in a simulation (*i.e.* either Vizdoom [23] or Habitat [59]), ARKitScenes provides raw RGB and depth of the scans and camera poses estimated using ARKit SLAM. In addition, whereas trajectories from the previous datasets are point-to-point, as typically done in navigation, the camera trajectories for ARKitScenes resembles a natural scan a of full indoor space. In our experiments we use a subset of 1K scans from ARKitScenes to train our models.

### 4.2 Reconstruction

We first validate the hypothesis that the optimization problem described in Eq. 2 can find latent codes $\mathbf{z}$ that are able reconstruct the trajectories in the empirical distribution in a satisfactory way. In Tab. 1 we report reconstruction performance of our model across all datasets. Fig. 4 shows reconstructions of random trajectories for each dataset. For all our experiments we set the dimension of $\mathbf{z}_{\text{scene}}$ and $\mathbf{z}_{\text{pose}}$ to 2048 and $\beta = 0.1$ unless otherwise stated. During training, we normalize camera poses for each trajectory so that the middle frame in a trajectory becomes the origin of the coordinate system. See appendix E for ablation experiments.

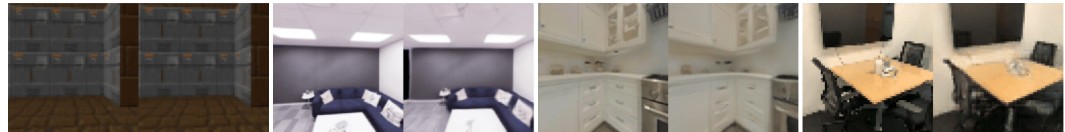

Figure 4: Qualitative reconstruction results of random trajectories on different datasets (one for each column): Vizdoom [23], Replica [64], VLN-CE [26]and ARKitScenes [1]. For each pair of images the left is ground-truth and right is reconstruction.

|  | #sc-#tr-#im | $l_1 \downarrow$ | PSNR $\uparrow$ | SSIM $\uparrow$ | Rot Err. $\downarrow$ | Trans. Err $\downarrow$ |
|---|---|---|---|---|---|---|
| Vizdoom [23] | 1-32-1k | 0.004 | 44.42 | 0.98 | 0.01 | 1.26 |
| Replica [64] | 18-100-1k | 0.006 | 38.86 | 0.99 | 0.03 | 0.01 |
| VLN-CE [26] | 90-3.6k-600k | 0.031 | 25.17 | 0.73 | 0.30 | 0.02 |
| ARKitScenes [1] | 300-1k-600k | 0.039 | 24.51 | 0.76 | 0.16 | 0.04 |

Table 1: Reconstruction results of the optimization process described in Eq. 2. The first column shows the number of scenes (#sc), trajectories (#tr) and images (#im) per dataset. Due to the large number of images on VLN-CE [26] and ARKitScenes [1] datasets we sample 10 random images per trajectory to compute the reconstruction metrics.

### 4.3 Interpolation

In addition, to evaluate the structure of the latent representation obtained from minimizing the optimization problem in Eq. 2, we show interpolation results between pairs of latents $(\mathbf{z}_i, \mathbf{z}_j)$ in

Fig. 5. To render images while interpolating the scene we place a fixed camera at the origin of the coordinate system. We observe a smooth transition of scenes in both geometry (walls, ceilings) and texture (stairs, carpets). More visualizations are included in the appendix H.

$\mathbf{z}_i$ $\longrightarrow$ $\mathbf{z}_j$

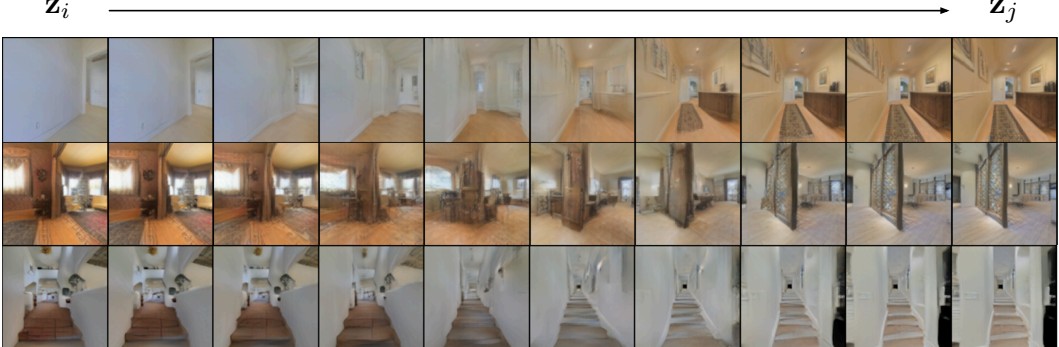

Figure 5: Interpolation of 3D scenes in latent space (*e.g.* interpolating the encoded radiance field) for the VLN-CE dataset [26]. Each row corresponds to a different interpolation path.

## 4.4 Unconditional generative modeling

Given latent representations $\mathbf{z} \in Z$ that can reconstruct samples $x \in X$ with high accuracy as shown in Sect. 4.2, we now evaluate the capacity of the prior $p_{\theta_p}(Z)$ to capture the empirical distribution $x \in \mathcal{X}$ by learning the distribution of latents $\mathbf{z}_i \in Z$. To do so we sample $\mathbf{z} \sim p_{\theta_p}(Z)$ by following the inference process in DDPMs, and then feed $\mathbf{z}$ through the decoder network, which results in trajectories of RGB images that are then used for evaluation. We compare our approach with the following baselines: GRAF [60], $\pi$-GAN [6] and GSN [8], where all models have access to ground-truth depth information during training. We sample 5k images from predicted and target distributions for each model and dataset and report both FID [16] and SwAV-FID [35] scores. We report quantitative results in Tab. 2, where we can see that GAUDI obtains state-of-the-art performance across all datasets and metrics. We attribute this performance improvement to the fact that GAUDI learns disentangled yet corresponding latents for radiance fields and camera poses, which is key when modeling scenes (see ablations in the appendix E). We note that to obtain these great empirical results GAUDI needs to simultaneously find latents with high reconstruction fidelity while also efficiently learning their distribution.

| | VizDoom [23] | | Replica [64] | | VLN-CE [26] | | ARKitScenes [1] | |
|---|---|---|---|---|---|---|---|---|
| | FID ↓ | SwAV-FID ↓ | FID ↓ | SwAV-FID ↓ | FID ↓ | SwAV-FID ↓ | FID ↓ | SwAV-FID ↓ |
| GRAF [60] | $47.50 \pm 2.13$ | $5.44 \pm 0.43$ | $65.37 \pm 1.64$ | $5.76 \pm 0.14$ | $90.43 \pm 4.83$ | $8.65 \pm 0.27$ | $87.06 \pm 9.99$ | $13.44 \pm 0.26$ |
| $\pi$-GAN [6] | $143.55 \pm 4.81$ | $15.26 \pm 0.15$ | $166.55 \pm 3.61$ | $13.17 \pm 0.20$ | $151.26 \pm 4.19674$ | $14.07 \pm 0.56$ | $134.80 \pm 10.60$ | $15.58 \pm 0.13$ |
| GSN [8] | $37.21 \pm 1.17$ | $4.56 \pm 0.19$ | $41.75 \pm 1.33$ | $4.14 \pm 0.02$ | $43.32 \pm 8.86$ | $6.19 \pm 0.49$ | $79.54 \pm 2.60$ | $10.21 \pm 0.15$ |
| GAUDI | $\mathbf{33.70 \pm 1.27}$ | $\mathbf{3.24 \pm 0.12}$ | $\mathbf{18.75 \pm 0.63}$ | $\mathbf{1.76 \pm 0.05}$ | $\mathbf{18.52 \pm 0.11}$ | $\mathbf{3.63 \pm 0.65}$ | $\mathbf{37.35 \pm 0.38}$ | $\mathbf{4.14 \pm 0.03}$ |

Table 2: Generative performance of state-of-the-art approaches for generative modelling of radiance fields on 4 scene datasets: Vizdoom [23], Replica [64], VLN-CE [26] and ARKitScenes [1], according to FID [16] and SwAV-FID [35] metrics.

In Fig. 6 we show samples from the unconditional distribution learnt by GAUDI for different datasets. We observe that GAUDI is able to generate diverse and realistic 3D scenes from the empirical distribution which can be rendered from the sampled camera poses.

## 4.5 Conditional Generative Modeling

GAUDI can also tackle conditional generative problems of the form $p(Z|Y)$, where a conditioning variable $y \in Y$ is given to condition $p(Z)$. For a given conditional inference problem we assume the existence of paired data [46, 11, 45]. As an example, for training a text-conditional model we assume the existence of pairs $\{\mathbf{z}_i, y_i\}$, where $\mathbf{z}_i$ is a latent scene representation and $y_i$ is its corresponding text prompt. In this section we show both quantitative and qualitative results for conditional inference

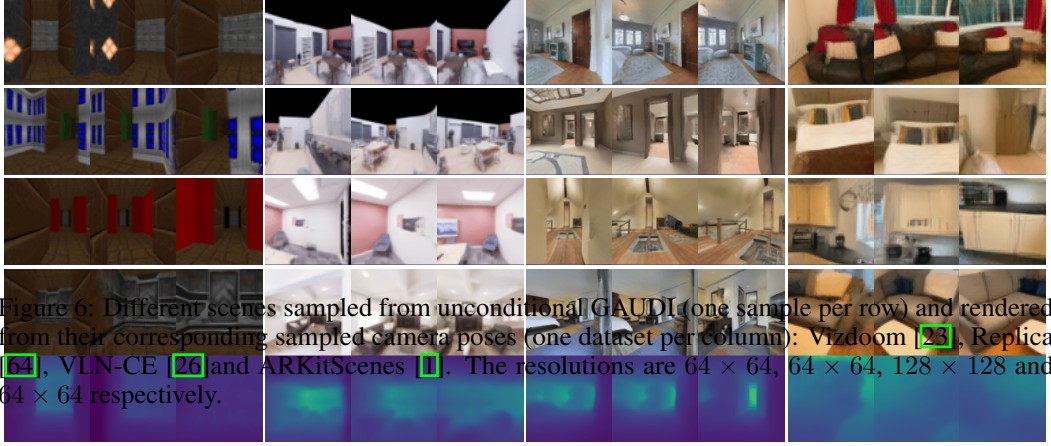

Figure 6: Different scenes sampled from unconditional GAUDI (one sample per row) and rendered from their corresponding sampled camera poses (one dataset per column): Vizdoom [23], Replica [64], VLN-CE [26] and ARKitScenes [1]. The resolutions are $64 \times 64$, $64 \times 64$, $128 \times 128$ and $64 \times 64$ respectively.

problems. The first conditioning variable we consider are textual descriptions of trajectories. Second, we consider a model where randomly sampled RGB images in a trajectory act as conditioning. Finally, we use a categorical variable that indicates the 3D environment (*i.e.* the particular indoor space) from which each trajectory was obtained. Tab. 3 shows quantitative results for the different conditional inference problems. Details on the implementation of the conditional DDPM are given in Sect. D

| Text Conditioning | | Image Conditioning | | Categorical Conditioning | | | Avg. $\Delta$ Per-Environment | |
|---|---|---|---|---|---|---|---|---|
| FID $\downarrow$ | SwAV-FID $\downarrow$ | FID $\downarrow$ | SwAV-FID $\downarrow$ | FID $\downarrow$ | SwAV-FID $\downarrow$ | | FID $\downarrow$ | SwAV-FID $\downarrow$ |
| 18.50 | 3.75 | 19.51 | 3.93 | 18.74 | 3.61 | | $-50.79$ | $-4.10$ |

Table 3: Quantitative results of Conditional Generative Modeling on VLN-CE [26] dataset. GAUDI is able to produce high-quality scene renderings with low FID and SwAV-FID scores. In the right table we show the difference in average *per-environment* FID score between the conditional and unconditional models.

### 4.5.1 Text Conditioning

We tackle the challenging task of training a text conditional model for 3D scene generation. We use the navigation text descriptions provided in VLN-CE [26] to condition our model. These text descriptions contain high level information about the scene as well as the navigation path (*i.e.* *"Walk out of the bedroom and into the living room"*, *"Exit the room through the swinging doors and then enter the bedroom"*). We employ a pre-trained RoBERTa-base [29] text encoder and use its intermediate representation to condition the diffusion model. Fig. 7 shows qualitative results of GAUDI for this task. To the best of our knowledge, this is the first model that allows for conditional 3D scene generation from text in an amortized manner (*i.e.* without distilling CLIP [44] through a costly optimization problem [19, 32]).

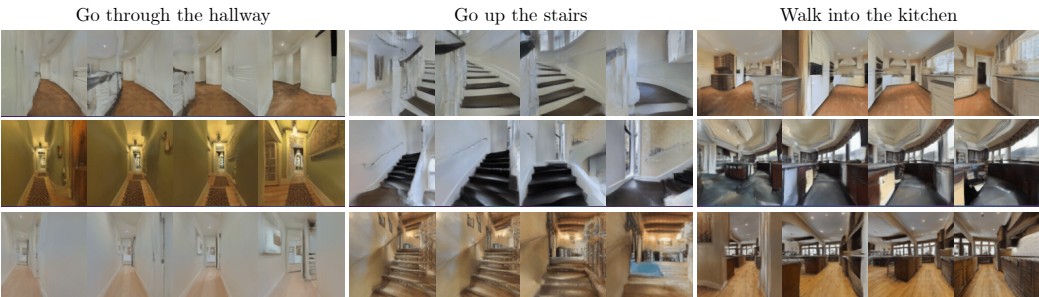

Figure 7: Text conditional 3D scene generation using GAUDI (one sample per row). Our model is able to capture the conditional distributions of scenes by generating multiple plausible scenes and camera paths that match the given text prompts.

### 4.5.2 Image Conditioning

We now analyze whether GAUDI is able to pick up information from the RGB images to predict a distribution over $Z$. In this experiment we randomly pick images in a trajectory $x \in X$ and use it as a conditioning variable $y$. For this experiment we use trajectories in the VLN-CE dataset [26]. During each training iteration we sample a random image for each trajectory $x$ and use it as a conditioning variable. We employ a pre-trained ResNet-18 [15] as an image encoder. During inference, the resulting conditional GAUDI model is able to sample radiance fields where the given image is observed from a stochastic viewpoint. In Fig. 8 we show samples from the model conditioned on different RGB images.

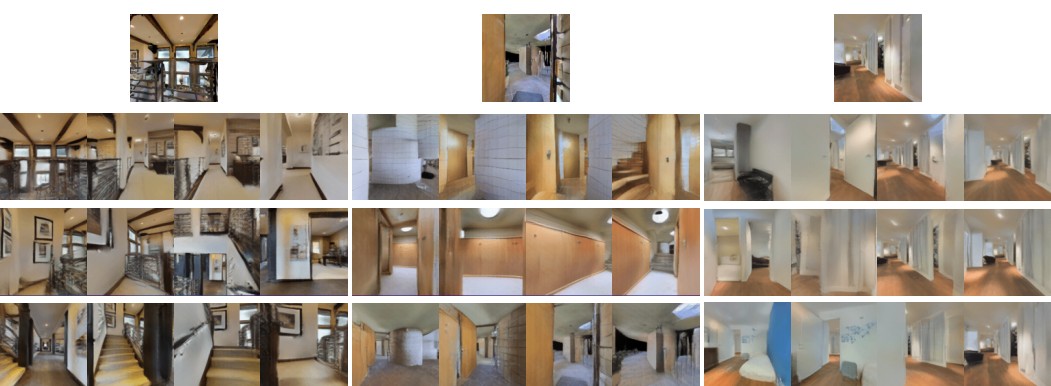

Figure 8: Image conditional 3D scene generation using GAUDI (one sample per row). Given a conditioned image (top row), our model is able to sample scenes where the same or contextually similar view is observed from a stochastic viewpoint.

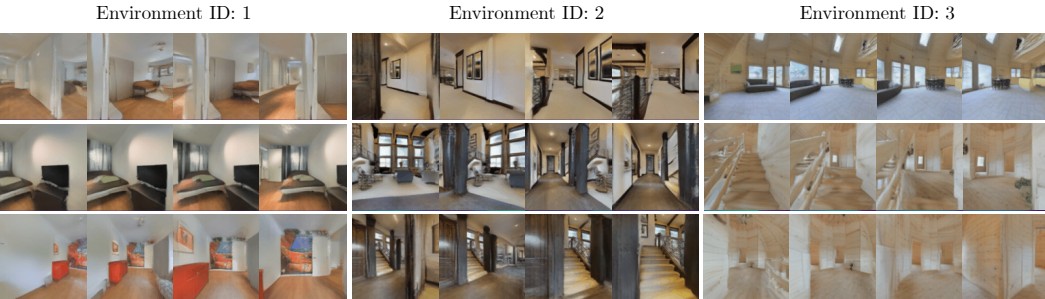

Figure 9: Samples from the GAUDI model conditioned on a categorical variable denoting the indoor scene (one sample per row).

### 4.5.3 Categorical Conditioning

Finally, we analyze how GAUDI performs when conditioned on a categorical variable that indicates the underlying 3D indoor environment in which each trajectory was recorded. We perform experiments in the VLN-CE [26] dataset, where we employ a trainable embedding layer to learn a representation for categorical variables indicating each environment. We compare the *per-environment* FID score of conditional model with its unconditional counterpart. This *per-enviroment* FID score is computed only on real images of the same indoor environment that the model is conditioned on. Our hypothesis is that if the model efficiently captures the information in the conditioning variable it should capture the environment specific distribution better than its unconditional counterpart trained on the same data. In Tab. 3 the last column shows difference (*e.g.* the $\Delta$) on the average *per-environment* FID score between the conditional and unconditional model on VLN-CE dataset. We observe that the conditional model consistently obtains a better FID score than the unconditional model across all indoor environments, resulting in a sharp reduction of average FID and SwAV-FID scores. In addition, in Fig. 9 we show samples from the model conditioned on a given categorical variable.

# 5 Conclusion

We have introduced GAUDI, a generative model that captures distributions of complex and realistic 3D scenes. GAUDI uses a scalable two-stage approach which first involves learning a latent representation that disentangles radiance fields and camera poses. The distribution of disentangled latent representations is then modeled with a powerful prior. Our model obtains state-of-the-art performance when compared with recent baselines across multiple 3D datasets and metrics. GAUDI can be used both for conditional and unconditional problems, and enabling new tasks like generating 3D scenes from text descriptions.

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
