# OpenReview forum: "GAUDI: A Neural Architect for Immersive 3D Scene Generation"
_NeurIPS.cc/2022/Conference — NeurIPS 2022 Accept_

### Official Review · Reviewer_TNJD · 2022-06-28

**Rating:** 6
**Confidence:** 3
**Soundness:** 3 good
**Presentation:** 3 good
**Contribution:** 3 good

**Summary:**

This paper generalizes previous NeRF-based generative models which focus on single objects to 3D scenes modeling where the canonical coordinate system may not exist. Instead of assuming the camera pose distribution to be shared across different scenes, the authors study the case that camera moves in the scene and thus forms a trajectory. They disentangle the modeling of camera pose and scene representation into different latent variables (and decoders) and learned these variables in an encoder-less method. After the latent variables are learned, the authors further learn a diffusion-based prior on them to enable sampling. Experimental results demonstrate the model's ability for reconstruction, interpolation, generative modeling for both unconditional and conditional cases.

**Questions:**

Please check the Strengths and Weaknesses part.

A minor question I woild like the authors to clearify here is for line 133, what does the "denoising recontrution objective" mean here. As I understand, the authors just use l2 reconstruction loss for image and l1 reconstruction loss for pose. Then what does "denoise" mean or where does the nose come from? Does it just mean you perturbed z with additive noise during training? Given the authors also use denoising diffusion probablistic model for prior learning in the next part, I think it might be better to make this more clear.

**Limitations:**

I think the authors adequately addressed the limitations and potential negative societal impact of their work.
But one potential limitation here might be whether the model can be scaled up to larger scenes like a trajectory accross multiple different rooms or an outdoor scene. The current model uses a single tri-plane representation for the whole scene, which means that the movements of the agent are limited to a predefined area. Then what if the agent moves outside the boundary？Simply enlarging the size of the predefined area size for larger areas or simply enlarging the size of the tri-plane representation seems not to be very efficient.

**Strengths And Weaknesses:**

For Strengths:

I think this paper study an important problem. Recently, the NeRF-based generative models show great potential on modeling single objects or simple scenes. On the other hand, modeling real-word, immersive 3D scenes where agent moves freely in the enviroment, is an important topic for many applications in robotics, VR/AR, but they are relatively under explored for NeRF-based generators. This paper proposes to solve this problem by disentangling the latent representations for camera pose and 3D enviroment. They also introduce the diffusion-based prior to enable sampling. Their model show good performance in both reconstruction and generation tasks. The paper is overall well-written and easy to follow.

Questions and potential weaknesses:
I have several concerns/questions regarding the experiments.
1. Novel view synthesis performance:  In 4.2 the authors show the reconstruction results. I assume these results are got from the trajectories/frames used to train the model. But how well does the model do for novel view synthesis? If during training we use a certain trajectory to get the scene representation $z_{scene}$ for a scene, then can this scene representation be generalized to model other trajectories in the same scene? (Or what will the PSNR/SSIM be if I use the learned scene representation to reconstruct a novel trajctory in the same room?) I think this is an important metrics to see whether the model just overfits the observed trajectories or it really builds a valid 3D representation.
2. Test on novel objects: Similar to 1, how well can the model do to reconstruct a novel scene that has not be seen during training?
3. For unconditional generation, the authors mainly demonstrate their model's ability through FID score, which is based on the quality of a single image. However, the model proposed here actually generates a trajectory in the scene. Would it be possible to compare the sample quality on the trajectory level (which, besides the quality of each frame, also consider the consistency among each frames)? A reference metrics here might be the FVD [1] score.
4. For conditional generation, why there are no baselines?

[1] Unterthiner, Thomas, et al. Towards Accurate Generative Models of Video: A New Metric & Challenges

---

> ### Author Response · Authors · 2022-08-02
> **Answers to TNJD**
>
> * “Novel view synthesis performance: In 4.2 the authors show the reconstruction results... But how well does the model do for novel view synthesis?“
>
>     * We thank the reviewer for bringing up this issue. We run an experiment to provide empirical evidence to support the fact the model doesn’t overfit to the camera poses seeing during training and that it allows for novel viewpoint synthesis. In this experiment tackle a model trained on VLN-CE [21] we perturb predicted camera poses with uniform noise both in translation (up to 50 cm) and orientation (up to 20 deg). We observe that while that there’s a small increase in FID metrics as we add noise, GAUDI still generates realistic images, outperforming previous approaches. We have included these analysis in section F of the appendix.
>         |                | No Noise       | 25 cm + 10 deg | 50 cm + 20 deg |
>         | -------------- | -------------- | -------------- | -------------- |
>         |FID             |    18.52       | 20.38          | 25.9           |
>         |SwaV-FID        |    3.63        | 4.01           | 4.68           |
>     * Additionally, in Sect. 4.5.2  we evaluate the performance of GAUDI for conditional inference of a scene given an image. This setting is the closest to novel view synthesis in a probabilistic formulation. We note that at its core view synthesis is not a deterministic problem but rather a stochastic one (eg. given an image, multiple completions of the scene are possible). We show that given an image, GAUDI can generate multiple trajectories that are consistent with the same underlying 3D scene. We encourage the reviewer to check the conditional inference results video in the appendix under “./cond_samples/cond_gen_raw_slides.mp4”.
> * “Test on novel objects: Similar to 1, how well can the model do to reconstruct a novel scene that has not be seen during training?”
>     * We thank the reviwer for this comment. GAUDI is a generative model, and as opposed to a 3d reconstruction method, the goal of GAUDI is to approximate the empirical distribution of scenes in the training set.  In order to generate novel samples, generative models learn high complex non-linear interpolations of samples in the training set. In our interpolation results in Fig. 5 and in the appendix we show that GAUDI is indeed able to generate novel scenes.
> * “For unconditional generation, the authors mainly demonstrate their model's ability through FID score, which is based on the quality of a single image. However, the model proposed here actually generates a trajectory in the scene. Would it be possible to compare the sample quality on the trajectory level (which, besides the quality of each frame, also consider the consistency among each frames)? A reference metrics here might be the FVD [1] score.”
>     * We thank the reviewer for pointing out using FVD as an additional metric to compute sample quality as a trajectory level. One consideration is that our trajectories are generally of different length (eg. different number of frames), which breaks the requirement for FVD and makes it not directly applicable.  We agree with the reviewer that trajectory level metrics can be an interesting metric to study.
> * “For conditional generation, why there are no baselines?”
>     * For conditional generation of 3D scenes there are no comparable baselines because GAUDI, as far as we know, is the first model to tackle this problem.
> * “What does the denoising recontrution objective mean in L133?"
>     * We agree with the reviewer that the terminology used in L133 "denoising recontrution objective" can cause confusion with the objective in DDPMs. The "denoising reconstruction objective" comes from the fact that while optimizing latents we add noise to the latents as explained in L137, this is completely separate from the next stage where we learn the generative model. We have gotten rid of the "denoising" term in L134 in the revised version of the manuscript.
> * “The current model uses a single tri-plane representation for the whole scene, which means that the movements of the agent are limited to a predefined area. Then what if the agent moves outside the boundary”
>     * We thank the reviewer for pointing this out, which we will include in our discussion of limitations. Indeed, the tri-plane representation defines the volume of physical space on which the radiance field is defined. Note that the cost of increasing the resolution of this representation scales quadratically with the volume of space, which already makes it a good candidate to model big indoor space with multiple rooms. To tackle the case in which one wants to model a infinitely big scenes, one could make both the camera pose and the radiance field representation $\mathbf{W}$ be a function of time step embedding $s$. This will allow for the radiance field to change as the camera moves. We have included this discussion in the revised version of the appendix L480-486.

---

> > ### Comment · Reviewer_TNJD · 2022-08-08
> > **Remaining concerns**
> >
> > I would like to thank the authors for their responses and extra efforts in the rebuttal phase. It solves some of my concerns, but not all.
> >
> > First, I'm afraid I can not agree with the authors that because their model is a generative model, it does not need to be justified on reconstructing novel scenes that follows the same distribution as the training data but are not in the training set. Just the opposite, for a generative model that successfully models an underlying distribution of different scenes, it should be able to reconstruct a new scene that follows the same distribution but hasn't been seen before. An extreme failure case might be that the model just memorizes all the training samples. In fact, generative models such as VAE[1] or EBM[2] usually report the metrics (bpd, NLL, mse) on a leave-out testing dataset instead of the training set.  I think showing interploation and generative results can to some extent demonstrate the model's capability, but that may not be enough.
> >
> > The second concern is about the trajectory-level metrcs. As the authors currently generate a trajectory instead of an single image, I think they should also show some quantitative results upon the quality of the whole sequences. This can demonstrate the consistency among the whole sequence and the performance of pose decoder (e.g. whether the pose decoder provides valid movements, and whether the movements form a continous trajectory or contain sudden pose jump). Although the authors show some successful qualitative results, quantitative results are more convincing to demonstrate whether these problems exist (it does not must be FVD, but I think some trajectory level judgements might be needed). Also, I see the authors say that they can not compute FVD because the sequences' lengths are different. But I'm curious why they can not just generate (or crop after generation) a fixed length of sequences and calculate the score.
> >
> >
> > [1] Vahdat, Arash, and Jan Kautz. "NVAE: A deep hierarchical variational autoencoder." Advances in Neural Information Processing Systems 33 (2020): 19667-19679.
> > [2] Pang, Bo, et al. "Learning latent space energy-based prior model." Advances in Neural Information Processing Systems 33 (2020): 21994-22008.

---

> > > ### Author Response · Authors · 2022-08-09
> > > **Thanks for engaging in discussion and additional FVD results.**
> > >
> > >
> > > We would like to thank the reviewer for engaging in discussions with us and for their comments. We address them in the following:
> > >
> > > * “I think showing interploation and generative results can to some extent demonstrate the model's capability, but that may not be enough.”
> > >
> > >     * We thank the reviewer for their comment and meant to just point out that the only way an unconditional generative model (eg. a model that has an objective of modeling p(scenes) ) can generalize to novel scenes is if it learns a good interpolation of the training samples (ie. encoding the appropriate factors of variation). In this direction, we want to highlight our results of generalization beyond the training set via interpolation in Fig 5 and appendix (Fig 11 and video). A deeper analysis like the ones performed for 2D image generative models would likely require a significantly larger training set to fill out the 3D space of variations of indoor scenes. We agree that this is a good direction for follow up works to GAUDI. In addition, we predict that the generalization/interpolation results obtained by GAUDI will improve as larger public 3D scene datasets become available. We believe this to be the case given two observations: (i) the empirical results observed on 2D models (e.g., DALL-E, Imagen, etc.) trained on large datasets, and (ii) our encouraging interpolations results.
> > > * “The second concern is about the trajectory-level metrcs. As the authors currently generate a trajectory instead of an single image, I think they should also show some quantitative results upon the quality of the whole sequences”
> > >     * We agree with the reviewer that this point is indeed interesting. In the last few days we have been thinking about possible ways of quantitatively evaluating the consistency in the trajectory, which is clearly visible in our qualitative results as pointed by the reviewer. Initially, our observations were the following: (i) None of the previous approaches (GRAF, pi-GAN, GSN) predict camera pose trajectories, which makes them non comparable with GAUDI in terms of FVD. (ii) Randomly cropping videos could introduce biases in FVD computation that might be encoded in future work that uses them as baseline.
> > >     * Following the reviewers recommendation we report the FVD score on 592 trajectories in VLN-CE which is  143.53. To put this number into context, DriveGAN[*4] a recent model for video prediction obtains an FVD score of 360.00 on Gibson (a dataset of indoor scenes that is very similar to VLN-CE). Computing the FVD score from the GT training trajectories to themselves results in a score of 43.09, this number serves as a lower bound in terms of FVD score that a model could obtain.
> > >
> > >
> > >
> > >     * In order to calculate the FVD score, we obtain clips of 20 equidistant frames from 592 trajectories predicted by our model as well as 592 randomly sampled ground truth videos. We then use the open repo: https://github.com/google-research/google-research/tree/master/frechet_video_distance to calculate the FVD score. Based on recent papers [*1,*2,*3,*4], the FVD score of 143.53 indicates that GAUDi is able to capture multi-view/temporal consistency as shown in the qualitative video samples provided in the appendix. We will keep thinking about different ways of qualitatively evaluating consistency and update the final appendix if we find a more suitable approach.
> > >     * [*1] Unterthiner, Thomas, et al. "Towards accurate generative models of video: A new metric & challenges." arXiv preprint arXiv:1812.01717 (2018).
> > >     * [*2] Castrejon, Lluis, Nicolas Ballas, and Aaron Courville. "Improved conditional vrnns for video prediction." Proceedings of the IEEE/CVF International Conference on Computer Vision. 2019.
> > >     * [*3] Menapace, Willi, et al. "Playable video generation." Proceedings of the IEEE/CVF Conference on Computer Vision and Pattern Recognition. 2021.
> > >     * [*4] Kim, Seung Wook, et al. "Drivegan: Towards a controllable high-quality neural simulation." Proceedings of the IEEE/CVF Conference on Computer Vision and Pattern Recognition. 2021.

---

### Official Review · Reviewer_si6w · 2022-07-11

**Rating:** 6
**Confidence:** 3
**Soundness:** 3 good
**Presentation:** 3 good
**Contribution:** 3 good

**Summary:**

This paper proposes a framework for 3D scene generation. The proposed framework enables the modeling of the scene distributions and scene-dependent camera distributions by learning priors from disentangled latent representations for radiance field and camera poses, which are obtained via optimizing a reconstruction objective over the training trajectories. Experimental results show that the proposed framework can be used for both unconditional and conditional generations. Extensive results are provided for ablation study to study the influence of key parameters and modules in the framework.

**Questions:**

1)	Since the results are demonstrated via short rendered videos under sampled camera path, it’s not that friendly to observe the diversity of the generated scenes. For example, the provided results for Replica in the supplementary video seem very similar. Are there any chances that authors can provide some visualizations of the underlying meshes or global layouts of the generated scenes?

2)	Is the proposed model able to generate longer camera paths (e.g, camera paths that are longer than the provided ones?)

3)	Can the proposed model guarantee the continuity and validness of the sampled camera poses where a new latent representation is sampled from the Gaussian distribution? If so, how does the proposed model achieve this? Is it achieved by the noise perturbations in latent optimization and the camera pose normalization?

4)	How long does it exactly take to generate a new scene?

5)	For text-based conditional generation, there are many artifacts observed within the provided results, what is the possible reason for this phenomenon?


**Limitations:**

The authors clearly state the limitations and potential negative impact of the proposed framework.

**Strengths And Weaknesses:**

The authors tackle the problem of 3D complex scene generation. For learning priors from a large amount of indoor scene observation trajectories, the authors propose to first learn a disentangled representation and then learn a generative model over the latent representations. The proposed method is technically sound. The writing and organization of this paper are good, and the authors provide sufficient experimental results in the manuscript and supplementary materials.
The technical part is not very easy to follow, especially for the conditional generation part. It would be better if more details can be given for better clarity.

Overall, the problem setting and the technical route of this work make sense to me. But there are still converns over the work in its current form. See the issue of the technical part above and more in the following.

---

> ### Author Response · Authors · 2022-08-02
> **Answers to si6w**
>
> * “The technical part is not very easy to follow, especially for the conditional generation part. It would be better if more details can be given for better clarity.”
>     * We thank the reviewer for this comment. We have clarified the conditional inference section in L237-246 of the revised paper and referenced detailed sections in the appendix.
>
> * “Since the results are demonstrated via short rendered videos under sampled camera path, it’s not that friendly to observe the diversity of the generated scenes. For example, the provided results for Replica in the supplementary video seem very similar. “
>     * In the supplementary material we provided videos with 64 samples generated from the prior for each dataset (Vizdoom, Replica, VLN-CE and ARKit). Indeed, since Replica only contains 18 scenes there are multiple trajectories that are sampled on the same underlying scene. Note that this is not the case for bigger datasets like VLN-CE or ARKit, for which videos are also provided in the supplementary.
> * “Is the proposed model able to generate longer camera paths (e.g, camera paths that are longer than the provided ones?)”
>     * We thank the reviewer for this comment. Since the camera pose decoder is queried with a “temporal embedding” $s$ that is continuous in the [-1, 1] interval,  our model can perform super-resolution of the camera paths, generating paths that are much longer than the ones seen during training. In addition, the training procedure for encoding trajectories is not limited to a specific length and can deal with arbitrary length trajectories.
> * “Can the proposed model guarantee the continuity and validness of the sampled camera poses where a new latent representation is sampled from the Gaussian distribution? If so, how does the proposed model achieve this? Is it achieved by the noise perturbations in latent optimization and the camera pose normalization?“
>     * The generative model learns the joint distribution of $z_scene$ and $z_pose$ latents . This means that every time we sample from the prior we get both a $z_scene$ and a $z_pose$ that we can decode into a continuous camera trajectory (via the combination of temporal embedding $s$ and $z_pose$) and use it to render RGB frames from the radiance field. The validity and continuity of the camera camera poses is learned from the training data.
> * “How long does it exactly take to generate a new scene?“
>     * We thank the reviewer for this comment, which have added to the revision of our manuscript L577-580.                Inference of the radiance field is amortize across the whole scene. This means that we only need to sample from the prior once and then we can render as many views of the scene as needed. Sampling from the prior takes 1.52s. Once we obtain the scene embedding, per frame rendering takes 1.6s at 128x128 resolution. These results are obtained on a single A100 NVIDIA GPU.
> * “For text-based conditional generation, there are many artifacts observed within the provided results, what is the possible reason for this phenomenon?“
>     * GAUDI is the first model to tackle text-based generation of unconstrained 3D scenes and as a result is not as fine-tuned as recent generative models for images like DALLE-2 or Imagen.  Additionally,  the size of the available 3d scene datasets is small relative to the data used to train DALLE-2 or Imagen. We expect this artifacts to be mitigated as the amount of training data increases. This is for two reasons: (i) a lot more 3D scene variety that can fill in the interpolation manifold. (ii) a larger amount of paired text-scene data will induce better mappings from text to scene.

---

### Official Review · Reviewer_gUke · 2022-07-11

**Rating:** 5
**Confidence:** 3
**Soundness:** 3 good
**Presentation:** 3 good
**Contribution:** 3 good

**Summary:**

This paper proposes a generative model for both indoor scenes and camera pose trajectories. The method can be trained as an uncondintional sampler or as a onditional one given a reference image/ a text prompt. At the core of the method, is a Denoising diffusion model based latent sampler, together with a scene generator, a trajactory generator, and a NeRF-based rendering pipeline. Compared with previous method, the proposed method scales well for indoor scenes, and generates reasonable results.

**Questions:**

1. As mentioned above, how robust is the method to camera pose registration errors? it seems the ARkit results is worse than the results from Replica.

2. How generalizable it is to use a DDPM to sample the latents? Would it still apply to other generative tasks where simple priors fail?

3. How robust is the method to succesfully train?

4. It seems that the generated results lack temporal coherency. Though that's quite common for NeRF based generators, is it possible to visualize the generated geometry/depth maps as well? It seems the NeRF model is view-dependent, would a view independent model be better at consistency?



**Limitations:**

The authors do not provide limitation discussions; it would be really helpful to have them.

**Strengths And Weaknesses:**

Originality:
The authors proposed a DDPM-based prior sampler together with trajectory and scene generation, which is a hard problem to tackel. The use of DDPM is well justified, as simple priors are limited in modeling different variations of camera poses and scene contents.

Quality:
The image generation quality is limited, but is understandable since the task is hard. The authors provided enough results to faithfully represent the quality of the method, which is good.

Significance:
I think this paper is of singinifcance especially to people interested in 3D aware video generation. But it should be noted that the method requires GT camera poses to train, and it seems the ARkit results are worse than Replica, potentially because the inaccuracies in camera calibration.

Clarity:
The paper is well written and easy to follow.

---

> ### Author Response · Authors · 2022-08-02
> **Answers to gUke**
>
> * “As mentioned above, how robust is the method to camera pose registration errors? it seems the ARkit results is worse than the results from Replica.“
>     * While our model still outperforms all the previous approaches on ARKit we agree with the reviewer that ARKit is the most challenging dataset due to a few factors: it contains real high-resolution scenes and it doesn’t provide ground truth depth or camera poses. In future work, we could backprop gradients from image reconstruction into the camera pose decoder network to fine-tune results [*1]. This can help mitigate some of the issues in ARKit.
> * “How generalizable it is to use a DDPM to sample the latents? Would it still apply to other generative tasks where simple priors fail?”
>     * Learning DDPMs in latent space is a very flexible approach for generative modeling [*3] specially for tasks where simple prior fails or are hard to encode like in radiance fields.
> * “How robust is the method to succesfully train?”
>     * GAUDI is extremely simple to train, this is due to the fact that our model boils down to a series of reconstruction objectives. First, we encode radiance fields and camera poses into latents by minimizing a reconstruction loss and then we train a DDPM model on this latent space, which again is series of denoising auto-encoders. Notably, when training the prior we didn't do grid search of hyper-parameters other than picking a fixed learning rate and a model size that can fit into a single Nvidia A100 GPU. GAUDI is much more robust than previous approach that adopt an adversarial formulation.
> * “It seems that the generated results lack temporal coherency. Though that's quite common for NeRF based generators, is it possible to visualize the generated geometry/depth maps as well? It seems the NeRF model is view-dependent, would a view independent model be better at consistency?”
>     * We thank the reviewer for this question. We believe the reviewer is referring to multi-view consistency as opposed to temporal consistency (since the scene is static and only the camera moves around).  The view inconsistency artifacts in the results experiment could be a result of performing the volumetric rendering in feature space and then use a convnet to upsample the feature to a desired resolution.  We adopt this commonly used trick from previous work [28, 6] to speed up training on large-scale datasets. In future work, we can consider tricks to improve the multi-view consistency as in StyleNerf [*2] to improve this kind of artifacts. Due to the fact that volumetric rendering happens in low-res feature space as in [28, 6], the resolution of the predicted depth is too small to be meaningful. Finally, by default we do remove the view dependent conditioning of NeRF in this paper. This is clarified in the revised manuscript in L126-127 and in Eq. (1).
> * Finally, we would like to kindly remind the reviewer that discussions on limitations and societal impact are provided in the supplementary material (as allowed in the submission policy).
>
> [*1] Yen-Chen, Lin, et al. "inerf: Inverting neural radiance fields for pose estimation." (IROS). IEEE, 2021.
>
> [*2] Gu, Jiatao, et al. "Stylenerf: A style-based 3d-aware generator for high-resolution image synthesis." arXiv preprint arXiv:2110.08985 (2021).
>
> [*3] Rombach, Robin, et al. "High-resolution image synthesis with latent diffusion models." Proceedings of the IEEE/CVF

---

> > ### Comment · Reviewer_gUke · 2022-08-08
> > **Thanks for the response**
> >
> > After reading all the comments and responses, I'm inclined to maintain my original rating as borderline accept.
> > I do believe, that this paper's merit outweights its flaws: The task of generating fly-through videos, with an option of text-prompt input, is a novel and interesting direction. The results shown in the paper are not perfect, but I do believe they deserve exposure.
> >
> > But I would like to stress that multiview consistency(or temporal consistency) is a major part of the problem, which is not being addressed in this paper. It is an important metric for evaluating the generation quality and should be somehow addressed in a comprehensive solution.
> >
> > Also, as the method's generation quality is directly correlated with camera pose and depth qualities, it might be hard to scale up to more natural datasets. Quality camera pose and depth maps are expensive to capture and are unlikly to scale up in the upcoming future. A more direct assessment on this issue would be more informative to the reader.
> >
> > That being said, I do think those drawbacks are acceptable comparing to the merit of this paper. Hence I would maintain my boarderline accept rating.

---

> > > ### Author Response · Authors · 2022-08-09
> > > **Thanks for engaging in discussion and additional FVD results.**
> > >
> > > We would like to thank the reviewer for engaging in discussions with us and for their comments. We address them in the following:
> > >
> > > * “But I would like to stress that multiview consistency(or temporal consistency) is a major part of the problem, which is not being addressed in this paper. It is an important metric for evaluating the generation quality and should be somehow addressed in a comprehensive solution.”
> > >     * We thank the reviewer for the comment. We just want to point out that our qualitative results show consistent scenes. We do believe that finding a suitable metric to measure this consistency is important future work. As a first step we provide FVD results for VLN-CE and will continue to work on this either for the final version of the appendix or future work.
> > >     * We report the FVD score on 592 trajectories in VLN-CE which is 143.53. To put this number into context, DriveGAN[*1] a recent model for video prediction obtains an FVD score of 360.00 on Gibson (a dataset of indoor scenes that is very similar to VLN-CE). Finally, computing the FVD score from the GT training trajectories to themselves results in a score of 43.09, this number serves as a lower bound in terms of FVD score that a model could obtain. The FVD score of 143.53 indicates that GAUDi is able to capture multi-view/temporal consistency as shown in the qualitative video samples provided in the appendix.
> > >     * [*1] Kim, Seung Wook, et al. "Drivegan: Towards a controllable high-quality neural simulation." Proceedings of the IEEE/CVF Conference on Computer Vision and Pattern Recognition. 2021.

---

### Official Review · Reviewer_KBe3 · 2022-07-12

**Rating:** 5
**Confidence:** 4
**Soundness:** 2 fair
**Presentation:** 2 fair
**Contribution:** 2 fair

**Summary:**

The paper proposed a method for the generative modeling of indoor scenes. The model contains a tri-plane based radiance field branch, and a camera trajectory branch. They are trained with indoor scene datasets that include RGBD images and ground truth camera trajectories.

The model employs an encoder-free architecture similar to DeepSDF [29]. Specifically, each individual scene in the dataset is associated with a learnable latent variable, which is optimized together with the model parameters during training.

The model is trained with RGB reconstruction loss, depth loss and loss between ground truth and predicted camera trajectories. After the auto-decoder model is trained, a diffusion model can be trained on top of the learned latents to enable sampling. The latent generative model can optionally be conditioned on text embeddings to achieve language-driven generation, provided that the dataset also contains language navigation.

**Questions:**

* There appears to be strong view inconsistencies in the interpolation video (in the Supp.) as the camera moves within the same scene (geometry or texture changes with camera pose). I wonder why is this happening?
* I wonder how does the proposed method (auto-decoder + latent DDPM) compare with GAN based methods in terms of synthesizing novel scenes? Does it have the tendency of memorizing the existing training scenes instead of generating novel ones?

**Limitations:**

Covered in the supplemental material.

**Strengths And Weaknesses:**

### Strengths
* Language-driven neural scene synthesis is a challenging problem which can lead to interesting applications.
* The paper shows that the encoder-free (auto-decoder) training objective used in DeepSDF [29] is also applicable to neural radiance fields. This significantly simplifies the training of a generalizable neural radiance field.
* The paper demonstrates that when the access to camera poses and depth maps is available, a 3D generative model producing higher quality images can be obtained with the use of stronger supervision signals.

### Weaknesses
* The comparison with previous methods, shown in Table 2 is not a fair comparison. The proposed model is trained with extra information from the dataset, such as ground truth camera poses, depths, and which images belong to the same scenes, while the baseline methods are trained with unposed images only. This important difference is not addressed in the paper. Instead, the paper choose to attribute the improvement to the learning of better latents (L224-228).
* Compared to previous methods, the proposed method puts scene-dependent constrains on the camera pose, limiting its use in arbitrary viewpoint synthesis. In fact, it appears that the model performed poorly on novel view/trajectories, as evident by Table 5 "GAUDI w. Random Pose" in the supplement material.
* The benefit of adding perturbation $\beta$ to the latent variables (L138) is not justified by the experiment. Supp. Table 4 shows that the reconstruction quality strictly decreases with larger $\beta$. What's worse, Supp. Table 5 also suggests that large $\beta$ affects generation performance while setting $\beta = 0$ leads to one of the best performing models.

---

> ### Author Response · Authors · 2022-08-02
> **Answers to KBe3**
>
> * “The comparison with previous methods, shown in Tab. 2 is not a fair comparison”
>     * All methods in Tab. 2 were trained using GT depth, following the protocol established in GSN [6]. This means that GT depth is concatenated to rgb along the channel dimension and fed to the discriminator (we explain this in L222 of the revised submission). In addition, in GAUDI we don’t assume knowledge about what images come from the same scene, the training set of trajectories are scene agnostic. Finally, we want to highlight that the camera poses for ARKit [1] dataset are estimated via an off-the-shelf SfM approach and GAUDI still outperforms previous approaches in this setting as show in Tab 2.
> * “The proposed method puts scene-dependent constrains on the camera pose, limiting its use in arbitrary viewpoint synthesis”
>     * We thank the reviewer for bringing up this point, which we have clarified in the revision L33-39. Notably,  scene-dependent constraints on the camera pose are needed to model unconstrained scenes with different layouts, like the ones we show in Fig. 3(b). Note how each scene defines different areas of navigable space (different dark dashed areas) where cameras can be placed. This is in opposition to generative models trained on datasets of single objects, like Shapenet, where the distribution of camera poses can be defined independently of the object, (eg. all cameras on the sphere are “valid cameras” independently of the object in Shapenet).
>     * In addition, we designed an experiment to show the performance in the arbitrary viewpoint synthesis setting. In this experiment we take a model trained on VLN-CE [21] and perturbed the camera poses sampled from the prior  with uniform noise both in translation (up to 50 cm) and orientation (up to 20 degrees). We observe that while that there’s an increase in FID metrics as we add noise, GAUDI still generates realistic images, outperforming previous approaches by a wide margin. This new experiment is included in the section F of the revised appendix.
>     * |                | No Noise       | 25 cm + 10 deg | 50 cm + 20 deg |
>         | -------------- | -------------- | -------------- | -------------- |
>         |FID             |    18.52       | 20.38          | 25.9           |
>         |SwaV-FID        |    3.63        | 4.01           | 4.68           |
>     * In Tab. 5 "GAUDI w. Random Pose", we show the result of completely breaking the dependence between scenes and valid camera poses, which results in a steep increase in FID as expected since views from non-valid camera poses are often rendered (eg. camera poses that are placed outside of the navigable area of a given scene).
> * “The benefit of adding perturbation β to the latent variables (L138) is not justified by the experiment.”
>     * We thank the reviewer for bringing up this point, which we have incorporated in L139-145 in the revised version. \beta can be interpreted as a weight that controls the smoothness of the distribution of latents. With \beta>0 we enforce a smoothing of the latent distribution enabling interpolation (as shown in Fig. 5 and appendix) at the cost of sacrificing fidelity, similar to comparing the latent space and reconstructions of AEs vs. VAEs for images.
> * “There appears to be strong view inconsistencies in the interpolation video”
>     * Interpolating radiance fields is a complex task since both geometry and appearance of the scene have to change jointly and consistently. In addition, as opposed to traditional stimuli like faces, scenes do not have a canonical orientation, which makes interpolation all the more challenging. To the best of our knowledge GAUDI is the first approach to show reasonable interpolation of scene-level radiance fields.  We expect that as the amount of training data for 3D scenes increases these artifacts will be mitigated. Future work can consider tricks to improve the multi-view consistency [*1]
> * “Memorization vs novel generation in GANs vs latent DDPMs”
>     * This is a fundamental question of generative modeling (for any data domain), the capacity of generative models to generate novel samples vs memorizing is an open problem.  From a training objective perspective a model that can perfectly memorize the training data distribution is a perfect generative model. During inference we rely on the functional iductive bias of the model to generate novel samples via non-linear interpolation of the training data manifold. Approaches like [*2, *3] show that training DDPMs in latent space for 2D images outperform GANs. We observed the same in GAUDI: our interpolated scenes are meaningful and the model outperforms previous GAN-based approaches.
>
> [*1] Gu, Jiatao, et al. "Stylenerf: A style-based 3d-aware generator for high-resolution image synthesis." arXiv:2110.08985
>
> [*2] Rombach, Robin, et al. "High-resolution image synthesis with latent diffusion models." CVPR 2022.
>
> [*3] Vahdat, et al. "Score-based generative modeling in latent space." NeurIPS 2021.

---

> > ### Author Response · Authors · 2022-08-07
> > **Thank you and open to further discussion if needed**
> >
> > We thank reviewer KEe3 for acknowledging receiving our rebuttal addressing reviewers comments and feedback. We are happy to discuss any additional points if needed.

---

> > ### Comment · Reviewer_KBe3 · 2022-08-08
> > **Thanks for the response**
> >
> > Dear authors,
> >
> > First, I would like to thank you for clarifying that the pi-GAN and GRAF baselines are both trained with ground truth depth. I am also glad to see that you have included this information in the revision. This is helpful as the settings used are not the standard settings in the original works.
> >
> > I am a little confused by the response *the training set of trajectories are scene agnostic*. If my understanding is correct, during training, each scene has a different set of camera trajectories specifically taylored to the scene, and these trajectories are modeled jointly with the scene aka *scene-dependent*. This is also mentioned in Supp. L600.
> >
> > As for the poor performance of random pose synthesis (Supp. Table 5), the author's theory of non-valid camera poses is partially convincing. However, it will be even more helpful if some visual evidences can be provided.
> >
> > I agree with the author that $\beta$ can potentially regularize the latent space. Nevertheless, its effect on the interpolation performance is never ablated.
> >
> > Finally, I agree that there is a potential that some imperfections of the proposed method, such as view-inconsistency, can be improved by the techniques from orthogonal works. However, since the proposed method is advertised as a 3D generative model, view-consistency is is a very important component. This is a concern shared by other reviewers as well (gUke; also related to TNJD's suggestion on using video metrics for evaluation).
> >
> > A suggestion: consider highlighting the updates in the revision to make the differences easier to spot.
> >
> > Best,
> > KBe3

---

> > > ### Author Response · Authors · 2022-08-09
> > > **Thanks for engaging in discussions and FVD results.**
> > >
> > > We would like to thank the reviewer for engaging in discussions with us and for their comments. We address them in the following:
> > >
> > > * “I am a little confused by the response the training set of trajectories are scene agnostic. If my understanding is correct, during training, each scene has a different set of camera trajectories specifically taylored to the scene, and these trajectories are modeled jointly with the scene aka scene-dependent. “
> > >     * We apologize for this misunderstanding. We meant that even though the trajectories depend on the layout of each 3D scene, when we train the unconditional model the trajectories cannot be mapped to a specific scene after being collected (eg. there’s no additional one-hot label that maps each trajectory to a specific 3D scene). Furthermore, our training sets for all datasets contain multiple trajectories for each 3D scene.
> > >
> > >
> > > * “Finally, I agree that there is a potential that some imperfections of the proposed method, such as view-inconsistency, can be improved by the techniques from orthogonal works. However, since the proposed method is advertised as a 3D generative model, view-consistency is is a very important component. This is a concern shared by other reviewers as well (gUke; also related to TNJD's suggestion on using video metrics for evaluation)”
> > >     * We thank the reviewer for this comment. We believe that finding suitable metrics for measuring consistency is going to be key for future work on scene generative models. We want to point out two observations:
> > >         * We could replace the convolutional upsampling network with a vanilla radiance field model that preserves multi-view consistency by design. This would be a drop-in replacement in the GAUDI framework, which will guarantee scene consistency at the cost of much higher training and inference runtimes. We opted for a convolutional upsampling network to reduce the computation cost. In follow up work on GAUDI we will focus on the trade-offs between training/inference costs and scene consistency.
> > >         * We have computed initial FVD scores for VLN-CE and will continue to think about suitable consistency metrics for the final version of the appendix. We report the FVD score on 592 trajectories in VLN-CE which is 143.53. To put this number into context, DriveGAN[*1] which is a recent model for video prediction obtains an FVD score of 360.00 on Gibson (a dataset of indoor scenes that is very similar to VLN-CE). Finally, computing the FVD score from the GT training trajectories to themselves results in a score of 43.09, this number serves as a lower bound in terms of FVD score that GAUDI could obtain. The FVD score of 143.53 indicates that GAUDi is able to capture multi-view/temporal consistency as shown in the qualitative video samples provided in the appendix.
> > >         * [*1] Kim, Seung Wook, et al. "Drivegan: Towards a controllable high-quality neural simulation." Proceedings of the IEEE/CVF Conference on Computer Vision and Pattern Recognition. 2021.

---

### Author Response · Authors · 2022-08-02
**Summary and changelog**

We thank the reviewers for all their thoughtful comments and insights, which have helped clarifying and strengthening the submission. Reviewers have highlighted the following strengths:

* si6w: “The authors tackle the problem of 3D complex scene generation. For learning priors from a large amount of indoor scene observation trajectories, the authors propose to first learn a disentangled representation and then learn a generative model over the latent representations. The proposed method is technically sound.”

* KBe3: “The paper shows that the encoder-free (auto-decoder) training objective used in DeepSDF [29] is also applicable to neural radiance fields. This significantly simplifies the training of a generalizable neural radiance field.”
* gUke: “The authors proposed a DDPM-based prior sampler together with trajectory and scene generation, which is a hard problem to tackel. The use of DDPM is well justified, as simple priors are limited in modeling different variations of camera poses and scene contents.”
* KBe3: “Language-driven neural scene synthesis is a challenging problem which can lead to interesting applications.”
* gUke: “The paper is well written and easy to follow.”

In addition, we have included explanations and experiments in our revised version of the paper and appendix to tackle reviewers questions and comments. The summary of the changes is as follows:

* Reviewer: KBe3 L33-39 (clarification): explanation of dependence between scenes and valid camera poses.
* Reviewer: gUke L126-128 and Eq. 1 (clarification): dropped orientation conditioning for radiance field.
* Reviewer: KBe3 L139-145 (clarification): clarifying the meaning of $\beta$.
* Reviewer: KBe3 L222 (clarification): all baselines use GT depth during training.
* Reviewer: si6w L237-246 (clarification): details on conditional inference problems.
* Reviewer: TNJD L480-486 (limitation): how to model infinitely big or “boundless” scenes.
* Reviewer: gUke L577-580 (clarification): inference time.
* Reviewer: KBe3/TNJD Appendix F (addition): experiments to test GAUDI for arbitrary viewpoint synthesis.

---

### Meta-Review · Area_Chair_Chga · 2022-08-24

**Recommendation:** Accept
**Confidence:** Certain

**Metareview:**

This paper proposes a framework to learn disentangled latent representation of radiance field and camera pose from trajectories of 3D scenes. The denoising diffusion probabilistic model can be further trained on the extracted latent representation for a conditioned or unconditioned generation. Experiments are conducted to validate the performance of the proposed method. The paper receives a total of 4 reviews. All reviewers lean to (borderline/weakly) accept the paper because of the novelty of the tasks, even though most of them raised concern about the view-consistency problem. AC recommends accepting the paper because the task of generating egocentric video, with an option of text-prompt input, is a novel and interesting direction, and this paper's merit outweighs its flaws.  AC urges the authors to improve their paper by taking into account all the feedback from the reviewers.

**Award:**

No

---

### Decision · Program_Chairs · 2022-09-14

Accept